# Biologically-Plausible Determinant Maximization Neural Networks for Blind Separation of Correlated Sources

**Bariscan Bozkurt**[1,2]    **Cengiz Pehlevan**[3]    **Alper T. Erdogan**[1,2]

[1]KUIS AI Center, Koc University, Turkey    [2]EEE Department, Koc University, Turkey
[3]John A. Paulson School of Engineering & Applied Sciences and Center for
Brain Science, Harvard University, Cambridge, 02138 MA, USA
`{bbozkurt15, alperdogan}@ku.edu.tr  cpehlevan@seas.harvard.edu`

## Abstract

Extraction of latent sources of complex stimuli is critical for making sense of the world. While the brain solves this blind source separation (BSS) problem continuously, its algorithms remain unknown. Previous work on biologically-plausible BSS algorithms assumed that observed signals are linear mixtures of statistically independent or uncorrelated sources, limiting the domain of applicability of these algorithms. To overcome this limitation, we propose novel biologically-plausible neural networks for the blind separation of potentially dependent/correlated sources. Differing from previous work, we assume some general geometric, not statistical, conditions on the source vectors allowing separation of potentially dependent/correlated sources. Concretely, we assume that the source vectors are sufficiently scattered in their domains which can be described by certain polytopes. Then, we consider recovery of these sources by the Det-Max criterion, which maximizes the determinant of the output correlation matrix to enforce a similar spread for the source estimates. Starting from this normative principle, and using a weighted similarity matching approach that enables arbitrary linear transformations adaptable by local learning rules, we derive two-layer biologically-plausible neural network algorithms that can separate mixtures into sources coming from a variety of source domains. We demonstrate that our algorithms outperform other biologically-plausible BSS algorithms on correlated source separation problems.

## 1  Introduction

Our brains constantly and effortlessly extract latent causes, or sources, of complex visual, auditory or olfactory stimuli sensed by sensory organs [1–11]. This extraction is mostly done without any instruction, in an unsupervised manner, making the process an instance of the blind source separation (BSS) problem [12, 13]. Indeed, visual and auditory cortical receptive fields were argued to be the result of performing BSS on natural images [1, 2] and sounds [4]. The wide-spread use of BSS in the brain suggests the existence of generic circuit motifs that perform this task [14]. Consequently, the literature on biologically-plausible neural network algorithms for BSS is growing [15–19].

Because BSS is an underdetermined inverse problem, BSS algorithms make generative assumptions on observations. In most instances of the biologically-plausible BSS algorithms, complex stimuli are assumed to be linear mixtures of latent sources. This assumption is particularly fruitful and is used to model, for example, natural images [1, 20], and responses of olfactory neurons to complex odorants [21–23]. However, linear mixing by itself is not sufficient for source identifiability; further assumptions are needed. Previous work on biologically-plausible algorithms for BSS of linear mixtures

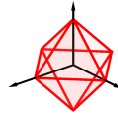 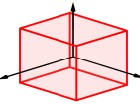 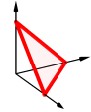 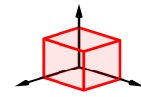 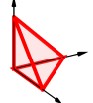

(a) $\mathcal{B}_{\ell_1}$ (sparse)  (b) $\mathcal{B}_{\ell_\infty}$ (anti-sparse)  (c) $\Delta$ (normalized nonnegative)  (d) $\mathcal{B}_{\ell_\infty,+}$ (nonnegative anti-sparse)  (e) $\mathcal{B}_{\ell_1,+}$ (nonnegative sparse)

Figure 1: Examples of source domains leading to identifiable generative models.

assumed sources to be statistically independent [17, 19, 24] or uncorrelated [16, 18]. However, these assumptions are very limiting when considering real data where sources can themselves be correlated.

In this paper, we address the limitation imposed by independence assumptions and provide biologically-plausible BSS neural networks that can separate potentially correlated sources. We achieve this by considering various general geometric identifiability conditions on sources instead of statistical assumptions like independence or uncorrelatedness. In particular, 1) we make natural assumptions on the domains of source vectors–like nonnegativity, sparsity, anti-sparsity or bounded-ness (Figure 1)–and 2) we assume that latent source vectors are sufficiently spread in their domain [25, 26]. Because these identifiability conditions are not stochastic in nature, our neural networks are able to separate both independent and dependent sources.

We derive our biologically-plausible algorithms from a normative principle. A common method for exploiting our geometric identifiability conditions is to disperse latent vector estimates across their presumed domain by maximizing the determinant of their sample correlation matrix, i.e., the Det-Max approach [25, 27–30]. Starting from a Det-Max objective function with constraints that specify the domain of source vectors, and using mathematical tools introduced for mapping optimization algorithms to adaptive Hebbian neural networks [18, 31, 32], we derive two-layered neural networks that can separate potentially correlated sources from their linear mixtures (Figure 2). These networks contain feedforward, recurrent and feedback synaptic connections updated via Hebbian or anti-Hebbian update rules. The domain of latent sources determines the structure of the output layer of the neural network (Figure 2, Table 1 and Appendix D).

In summary, our main contributions in this article are the following:

- We propose a normative framework for generating biologically plausible neural networks that are capable of separating correlated sources from their mixtures by deriving them from a Det-Max objective function subject to source domain constraints.
- Our framework can handle infinitely many source types by exploiting their source domain topology.
- We demonstrate the performance of our networks in simulations with synthetic and realistic data.

## 1.1 Other related work

Several algorithms for separation of linearly mixed and correlated sources have been proposed outside the domain of biologically-plausible BSS. These algorithms make other forms of assumptions on the latent sources. Nonnegative matrix factorization (NMF) assumes that the latent vectors are nonnegative [13, 33–35]. Simplex structured matrix factorization (SSMF) assumes that the latent vectors are members of the unit-simplex [25, 36, 37]. Sparse component analysis (SCA) often assumes that the latent vectors lie in the unity $\ell_1$-norm-ball [30, 38–42]. Antisparse bounded component analysis (BCA) assumes latent vectors are in the $\ell_\infty$-norm-ball [28, 29, 43]. Recently introduced polytopic matrix factorization (PMF) extends the identifiability-enabling domains to infinitely many polytopes obeying a particular symmetry restriction [26, 44, 45].

The mapping of optimization algorithms to biologically-plausible neural networks have been formal-ized in the similarity matching framework [31, 32, 46, 47]. Several BSS algorithms were proposed within this framework: 1) Nonnegative Similarity Matching (NSM) [16, 48] separates linear mix-tures of uncorrelated nonnegative sources, 2) [19] separates independent sources, and 3) Bounded Similarity Matching (BSM) separates uncorrelated anti-sparse bounded sources from $\ell_\infty$-norm-ball

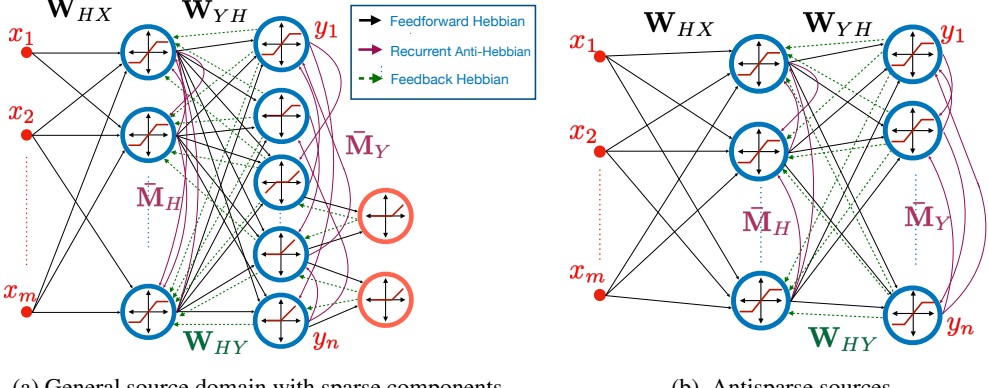

(a) General source domain with sparse components      (b) Antisparse sources

Figure 2: Det-Max WSM neural network for blind source separation. The network takes a mixed input $\mathbf{x}$ and produces latent components $\mathbf{y}$ at the output. The output layer depends on the choice of source domain. Mutually sparse components are connected by inhibitory neurons at the output layer.

[18]. BSM introduced a weighted inner product-based similarity criterion, referred to as the weighted similarity matching (WSM). Compared to these algorithm, the neural network algorithms we propose in this article 1) cover more general source domains, 2) handle potentially correlated sources, 3) use a two-layer WSM architecture (relative to single layer WSM architecture of BSM, which is not capable of generating arbitrary linear transformations) and 4) offer a general framework for neural-network-based optimization of the Det-Max criterion.

## 2 Problem statement

### 2.1 Sources

We assume that there are $n$ real-valued sources, represented by the vector $\mathbf{s} \in \mathcal{P}$, where $\mathcal{P}$ is a particular subset of $\mathbb{R}^n$. Our algorithms will address a wide range of source domains. We list some examples before giving a more general criterion:

- *Bounded sparse sources*: A natural convex domain choice for sparse sources is the unit $\ell_1$ norm ball $\mathcal{B}_{\ell_1} = \{\mathbf{s} \ | \ \|\mathbf{s}\|_1 \leq 1\}$ (Figure 1.(a)). The use of $\ell_1$-norm as a convex (non)sparsity measure has been quite successful with various applications including sparse dictionary learning/component analysis [30, 39, 41, 49, 50] and modeling of V1 receptive fields [2].
- *Bounded anti-sparse sources*: A common domain choice for anti-sparse sources is the unit $\ell_\infty$-norm-ball: $\mathcal{B}_{\ell_\infty} = \{\mathbf{s} \ | \ \|\mathbf{s}\|_\infty \leq 1\}$ (Figure 1.(b)). If vectors drawn from $\mathcal{B}_{\ell_\infty}$ are well-spread inside this set, some samples would contain near-peak magnitude values simultaneously at all their components. The potential equal spreading of values among the components justifies the term "anti-sparse" [51] or "democratic" [52] component representations. This choice is well-suited for both applications in natural images and digital communication constellations [28, 43].
- *Normalized nonnegative sources*: Simplex structured matrix factorization [25, 36, 37] uses the unit simplex [35, 53] $\Delta = \{\mathbf{s} \ | \ \mathbf{s} \geq 0, \mathbf{1}^T \mathbf{s} = 1\}$ (Figure 1.(c)) as the source domain. Nonnegativity of sources naturally arises in biological context, for example in demixing olfactory mixtures [54].
- *Nonnegative bounded anti-sparse sources*: A non-degenerate polytopic choice of the nonnegative sources can be obtained through the combination of anti-sparseness and nonnegativity constraints. This corresponds to the intersection of $\mathcal{B}_{\ell_\infty}$ with the nonnegative orthant $\mathbb{R}_+^n$, represented as $\mathcal{B}_{\ell_\infty,+} = \mathcal{B}_{\ell_\infty} \cap \mathbb{R}_+^n$ [26] (Figure 1.(d)).
- *Nonnegative bounded sparse sources*: Another polytopic choice for nonnegative sources can be obtained through combination of the sparsity and nonnegativity constraints which yields the intersection of $\mathcal{B}_{\ell_1}$ with the nonnegative orthant $\mathbb{R}_+$, [26]: $\mathcal{B}_{\ell_1,+} = \mathcal{B}_{\ell_1} \cap \mathbb{R}_+^n$ (Figure 1.(e)).

Except the unit simplex $\Delta$, all the examples above are examples of an infinite set of identifiable polytopes whose symmetry groups are restricted to the combinations of component permutations and sign alterations as formalized in PMF framework for BSS [44]. Further, in-

stead of a homogeneous choice of features, such as sparsity and nonnegativity, globally imposed on all elements of the component vector, we can assign these attributes at the subvector level and still obtain identifiable polytopes. For example, the reference [26] provides the set $\mathcal{P}_{ex} = \left\{ \mathbf{s} \in \mathbb{R}^3 \,\middle|\, s_1, s_2 \in [-1, 1], s_3 \in [0, 1], \left\| \begin{bmatrix} s_1 \\ s_2 \end{bmatrix} \right\|_1 \leq 1, \left\| \begin{bmatrix} s_2 \\ s_3 \end{bmatrix} \right\|_1 \leq 1 \right\}$, as a simple illustration of such polytopes with heterogeneous structure where $s_3$ is nonnegative, $s_1, s_2$ are signed, and $\begin{bmatrix} s_1 & s_2 \end{bmatrix}^T, \begin{bmatrix} s_2 & s_3 \end{bmatrix}^T$ are sparse subvectors, while sparsity is not globally imposed. In this article, we concentrate on particular source domains including the unit simplex, and the subset of identifiable polytopes for which the attributes such as sparsity and nonnegativity are defined at the subvector level in the general form

$$\mathcal{P} = \left\{ \mathbf{s} \in \mathbb{R}^n \,\middle|\, s_i \in [-1, 1] \, \forall i \in \mathcal{I}_s, \, s_i \in [0, 1] \, \forall i \in \mathcal{I}_+, \, \|\mathbf{s}_{\mathcal{J}_k}\|_1 \leq 1, \, \mathcal{J}_k \subseteq \mathbb{Z}_n, \, k \in \mathbb{Z}_L \right\}, \quad (1)$$

where $\mathcal{I}_+ \subseteq \mathcal{Z}_n$ is the index set for nonnegative sources, and $I_s$ is its complement, $\mathbf{s}_{\mathcal{J}_k}$ is the subvector constructed from the elements with indices in $\mathcal{J}_k$, and $L$ is the number of sparsity constraints imposed in the subvector level.

The Det-Max criterion for BSS is based on the assumption that the source samples are well-spread in their presumed domain. The references [55] and [26] provide precise conditions on the scattering of source samples which guarantee their identifiability for the unit simplex and polytopes, respectively. Appendix A provides a brief summary of these conditions.

We emphasize that our assumptions about the sources are deterministic. Therefore, our proposed algorithms do not exploit any stochastic assumptions such as independence or uncorrelatedness, and can separate both independent and dependent (potentially correlated) sources.

## 2.2 Mixing

The sources $\mathbf{s}_t$ are mixed through a mixing matrix $\mathbf{A} \in \mathbb{R}^{m \times n}$.

$$\mathbf{x}_t = \mathbf{A}\mathbf{s}_t, \qquad t \in \mathbb{Z}. \tag{2}$$

We only consider the (over)determined case with $m \geq n$ and assume that the mixing matrix is full-rank. While we consider noiseless mixtures to achieve perfect separability, the optimization setting proposed for the online algorithm features a particular objective function that safeguards against potential noise presence. We use $\mathbf{S}(t) = \begin{bmatrix} \mathbf{s}_1 & \dots & \mathbf{s}_t \end{bmatrix} \in \mathbb{R}^{n \times t}$ and $\mathbf{X}(t) = \begin{bmatrix} \mathbf{x}_1 & \dots & \mathbf{x}_t \end{bmatrix} \in \mathbb{R}^{m \times t}$ to represent data snapshot matrices, at time $t$, for sources and mixtures, respectively.

## 2.3 Separation

The goal of the source separation is to obtain an estimate of $\mathbf{S}(t)$ from the mixture measurements $\mathbf{X}(t)$ when the mixing matrix $\mathbf{A}$ is unknown. We use the notation $\mathbf{y}_t$ to refer to source estimates, which are linear transformations of observations, i.e., $\mathbf{y}_i = \mathbf{W}\mathbf{x}_i$, where $\mathbf{W} \in \mathbb{R}^{n \times m}$. We define $\mathbf{Y}(t) = \begin{bmatrix} \mathbf{y}_1 & \mathbf{y}_2 & \dots & \mathbf{y}_t \end{bmatrix} \in \mathbb{R}^{n \times t}$ as the output snapshot matrix. "Ideal separation" is defined as the condition where the outputs are scaled and permuted versions of original sources, i.e., they satisfy $\mathbf{y}_t = \mathbf{P}\mathbf{\Lambda}\mathbf{s}_t$, where $\mathbf{P}$ is a permutation matrix, and $\mathbf{\Lambda}$ is a full rank diagonal matrix.

# 3 Determinant maximization based blind source separation

Among several alternative solution methods for the BSS problem, the determinant-maximization (Det-Max) criterion has been proposed within the NMF, BCA, and PMF frameworks, [26–28, 30, 35, 44]. Here, the separator is trained to maximize the (log)-determinant of the sample correlation matrix for the separator outputs, $J(\mathbf{W}) = \log(\det(\hat{\mathbf{R}}_y(t)))$, where $\hat{\mathbf{R}}_y(t)$ is the sample correlation matrix $\hat{\mathbf{R}}_y(t) = \frac{1}{t} \sum_{i=1}^{t} \mathbf{y}_i \mathbf{y}_i^T = \frac{1}{t} \mathbf{Y}(t) \mathbf{Y}(t)^T$. Further, during the training process, the separator outputs are constrained to lie inside the presumed source domain, i.e. $\mathcal{P}$. As a result, we can pose the corresponding optimization problem as [26, 35]

$$\underset{\mathbf{Y}(t)}{\text{maximize}} \quad \log(\det(\mathbf{Y}(t)\mathbf{Y}(t)^T)) \tag{3a}$$

$$\text{subject to} \quad \mathbf{y}_i \in \mathcal{P}, i = 1, \dots, t, \tag{3b}$$

where we ignored the constant $\frac{1}{t}$ term. Here, the determinant of the correlation matrix acts as a spread measure for the output samples. If the original source samples $\{\mathbf{s}_1, \ldots, \mathbf{s}_t\}$ are sufficiently scattered inside the source domain $\mathcal{P}$, as described in Section 2.1 and Appendix A, then the global solution of this optimization can be shown to achieve perfect separation [26, 35, 55].

## 4    An alternative optimization formulation of determinant-maximization based on weighted similarity matching

Here, we reformulate the Det-Max problem 3 described above in a way that allows derivation of a biologically-plausible neural network for the linear BSS setup in Section 2. Our formulation applies to all source types discusses in 2.1.

We propose the following optimization problem:

$$\underset{\substack{\mathbf{Y}(t),\mathbf{H}(t),D_{1,11}(t),\ldots D_{1,nn}(t),\mathbf{D}_1(t) \\ D_{2,11}(t),\ldots D_{2,nn}(t),\mathbf{D}_2(t)}}{\text{minimize}} \quad \sum_{i=1}^{n} \log(D_{1,ii}(t)) + \sum_{i=1}^{n} \log(D_{2,ii}(t)) \tag{4a}$$

$$\text{subject to} \qquad \mathbf{X}(t)^T \mathbf{X}(t) - \mathbf{H}(t)^T \mathbf{D}_1(t) \mathbf{H}(t) = 0, \tag{4b}$$

$$\mathbf{H}(t)^T \mathbf{H}(t) - \mathbf{Y}(t)^T \mathbf{D}_2(t) \mathbf{Y}(t) = 0, \tag{4c}$$

$$\mathbf{y}_i \in \mathcal{P}, i = 1, \ldots, n, \tag{4d}$$

$$\mathbf{D}_l(t) = \text{diag}(D_{l,11}(t), \ldots, D_{l,nn}(t)), \quad l = 1, 2, \tag{4e}$$

$$D_{l,11}(t), D_{l,22}(t), \ldots, D_{l,nn}(t) > 0, \quad l = 1, 2 \tag{4f}$$

Here, $\mathbf{X}(t) \in \mathbb{R}^{m \times t}$ is the matrix containing input (mixture) vectors, $\mathbf{Y}(t) \in \mathbb{R}^{n \times t}$ is the matrix containing output vectors, $\mathbf{H}(t) \in \mathbb{R}^{n \times t}$ is a slack variable containing an intermediate signal $\{\mathbf{h}_i \in \mathbb{R}^n, i = 1, \ldots, t\}$, corresponding to the hidden layer of the neural network implementation in Section 5, in its columns $\mathbf{H}(t) = \begin{bmatrix} \mathbf{h}_1 & \mathbf{h}_2 & \ldots & \mathbf{h}_t \end{bmatrix}$. $D_{l,11}(t), D_{l,22}(t), \ldots, D_{l,nn}(t)$ for $l = 1, 2$ are nonnegative slack variables to be described below, and $\mathbf{D}_l$ is the diagonal matrix containing weights $D_{l,ii}$ for $i = 1, \ldots, n$ and $l = 1, 2$. The constraint (4d) ensures that the outputs lie in the presumed domain of sources.

This problem is related to the weighted similarity matching (WSM) objective introduced in [18]. Constraints (4b) and (4c) define two separate WSM conditions. In particular, the equality constraint in (4b) is a WSM constraint between inputs and the intermediate signal $\mathbf{H}(t)$. This constraint imposes that the pairwise weighted correlations of the signal $\{\mathbf{h}_i, i = 1, \ldots, t\}$ are the same as correlations among the elements of the input signal $\{\mathbf{x}_i, i = 1, \ldots, t\}$, i.e., $\mathbf{x}_i^T \mathbf{x}_j = \mathbf{h}_i^T \mathbf{D}_1(t) \mathbf{h}_j$, $\forall i, j \in \{1, \ldots, t\}$. $D_{1,11}(t), D_{1,22}(t), \ldots, D_{1,nn}(t)$ correspond to inner product weights used in these equalities. Similarly, the equality constraint in (4c) defines a WSM constraint between the intermediate signal and outputs. This equality can be written as $\mathbf{h}_i^T \mathbf{h}_j = \mathbf{y}_i^T \mathbf{D}_2(t) \mathbf{y}_j$, $i, j \in \{1, \ldots, t\}$, and $D_{2,11}(t), D_{2,22}(t), \ldots, D_{2,nn}(t)$ correspond to the inner product weights used in these equalities. The optimization involves minimizing the logarithm of the determinant of the weighting matrices.

Now we state the relation between our WSM-based objective and the original Det-Max criterion (3).

**Theorem 1.** *If $\mathbf{X}(t)$ is full column-rank, then global optimal $\mathbf{Y}(t)$ solutions of (3) and (4) coincide.*

*Proof of Theorem 1.* See Appendix B for the proof. The proof relies on a lemma that states that the optimization constraints enforce inputs and outputs to be related by an arbitrary linear transformation. □

## 5    Biologically-plausible neural networks for WSM-based BSS

The optimization problems we considered so far were in an offline setting, where all inputs are observed together and all outputs are produced together. However, biology operates in an online fashion, observing an input and producing the corresponding output, before seeing the next input. Therefore, in this section, we first introduce an online version of the batch WSM-problem (4). Then we show that the corresponding gradient descent algorithm leads to a two-layer neural network with biologically-plausible local update rules.

## 5.1 Online optimization setting for WSM-based BSS

We first propose an online extension of WSM-based BSS (4). In the online setting, past outputs cannot be altered, but past inputs and outputs still carry valuable information about solving the BSS problem. We will write down an optimization problem whose goal is to produce the sources $\mathbf{y}_t$ given a mixture $\mathbf{x}_t$, while exploiting information from all the fixed previous inputs and outputs.

We first introduce our notation. We consider exponential weighting of the signals as a recipe for dynamical adjustment to potential nonstationarity in the data. We define the weighted input data snapshot matrix by time $t$ as, $\boldsymbol{\mathcal{X}}(t) = \begin{bmatrix} \gamma^{t-1}\mathbf{x}_1 & \dots & \gamma\mathbf{x}_{t-1} & \mathbf{x}_t \end{bmatrix} = \mathbf{X}(t)\boldsymbol{\Gamma}(t)$, where $\gamma$ is the forgetting factor and $\boldsymbol{\Gamma}(t) = \mathrm{diag}(\gamma^{t-1}, \dots, \gamma, 1)$. The exponential weighting emphasizes recent mixtures by reducing the impact of past samples. Similarly, we define the corresponding weighted output snapshot matrix for output as $\boldsymbol{\mathcal{Y}}(t) = \begin{bmatrix} \gamma^{t-1}\mathbf{y}_1 & \dots & \gamma\mathbf{y}_{t-1} & \mathbf{y}_t \end{bmatrix} = \mathbf{Y}(t)\boldsymbol{\Gamma}(t)$, and the hidden layer vectors as $\boldsymbol{\mathcal{H}}(t) = \begin{bmatrix} \gamma^{t-1}\mathbf{h}_1 & \dots & \gamma\mathbf{h}_{t-1} & \mathbf{h}_t \end{bmatrix} = \mathbf{H}(t)\boldsymbol{\Gamma}(t)$. We further define $\tau = \lim_{t\to\infty} = \sum_{k=0}^{t-1}\gamma^{2k} = \frac{1}{1-\gamma^2}$ as a measure of the effective time window length for sample correlation calculations based on the exponential weights.

In order to derive an online cost function, we first converted equality constraints in (4b) and (4c) to similarity matching cost functions $J_1(\mathbf{H}(t), \mathbf{D}_1(t)) = \frac{1}{2\tau^2}\|\boldsymbol{\mathcal{X}}(t)^T\boldsymbol{\mathcal{X}}(t) - \boldsymbol{\mathcal{H}}(t)^T\mathbf{D}_1(t)\boldsymbol{\mathcal{H}}(t)\|_F^2$, $J_2(\mathbf{H}(t), \mathbf{D}_2(t), \mathbf{Y}(t)) = \frac{1}{2\tau^2}\|\boldsymbol{\mathcal{H}}(t)^T\boldsymbol{\mathcal{H}}(t) - \boldsymbol{\mathcal{Y}}(t)^T\mathbf{D}_2(t)\boldsymbol{\mathcal{Y}}(t)\|_F^2$. Then, a weighted combination of similarity matching costs and the objective function in (4a) yields the final cost function

$$
\begin{aligned}
\mathcal{J}(\mathbf{H}(t), \mathbf{D}_1(t), \mathbf{D}_2(t), \mathbf{Y}(t)) = &\ \lambda_{SM}[\beta J_1(\mathbf{H}(t), \mathbf{D}_1(t)) + (1-\beta)J_2(\mathbf{H}(t), \mathbf{D}_2(t), \mathbf{Y}(t))] \\
&+ (1-\lambda_{SM})[\sum_{k=1}^n \log(D_{1,kk}(t)) + \sum_{k=1}^n \log(D_{2,kk}(t))]. \quad (5)
\end{aligned}
$$

Here, $\beta \in [0,1]$ and $\lambda_{SM} \in [0,1]$ are parameters that convexly combine similarity matching costs and the objective function. Finally, we can state the online optimization problem for determining the current output $\mathbf{y}_t$, the corresponding hidden state $\mathbf{h}_t$ and for updating the gain parameters $\mathbf{D}_l(t)$ for $l = 1, 2$, as

$$
\underset{\substack{\mathbf{y}_t, \mathbf{h}_t, D_{1,11}(t), \dots D_{1,nn}(t), \mathbf{D}_1(t) \\ D_{2,11}(t), \dots D_{2,nn}(t), \mathbf{D}_2(t)}}{\text{minimize}} \quad \mathcal{J}(\mathbf{H}(t), \mathbf{D}_1(t), \mathbf{D}_2(t), \mathbf{Y}(t)) \tag{6a}
$$

$$
\text{subject to} \quad\quad\quad\quad\quad\quad\quad\quad \mathbf{y}_t \in \mathcal{P}, \tag{6b}
$$
$$
\mathbf{D}_l(t) = \mathrm{diag}(D_{l,11}(t), \dots, D_{l,nn}(t)), \quad l = 1, 2, \tag{6c}
$$
$$
D_{l,11}(t), D_{l,22}(t), \dots, D_{l,nn}(t) > 0, \quad l = 1, 2 \tag{6d}
$$

As shown in Appendix C.1, part of $\mathcal{J}$ that depends on $\mathbf{h}_t$ and $\mathbf{y}_t$ can be written as

$$
\begin{aligned}
C(\mathbf{h}_t, \mathbf{y}_t) = &\ 2\mathbf{h}_t^T\mathbf{D}_1\mathbf{M}_H(t)\mathbf{D}_1(t)\mathbf{h}_t - 4\mathbf{h}_t^T\mathbf{D}_1(t)\mathbf{W}_{HX}(t)\mathbf{x}_t \\
&+ 2\mathbf{y}_t^T\mathbf{D}_2(t)\mathbf{M}_Y(t)\mathbf{D}_2(t)\mathbf{y}_t - 4\mathbf{y}_t^T\mathbf{D}_2(t)\mathbf{W}_{YH}(t)\mathbf{h}_t + 2\mathbf{h}_t^T\mathbf{M}_H(t)\mathbf{h}_t, \quad (7)
\end{aligned}
$$

where the dependence on past inputs and outputs appear in the weighted correlation matrices:

$$
\begin{aligned}
\mathbf{M}_H(t) = \frac{1}{\tau}\sum_{k=1}^{t-1}(\gamma^2)^{t-1-k}\mathbf{h}_k\mathbf{h}_k^T, \quad & \mathbf{W}_{HX}(t) = \frac{1}{\tau}\sum_{k=1}^{t-1}(\gamma^2)^{t-1-k}\mathbf{h}_k\mathbf{x}_k^T, \\
\mathbf{W}_{YH}(t) = \frac{1}{\tau}\sum_{k=1}^{t-1}(\gamma^2)^{t-1-k}\mathbf{y}_k\mathbf{h}_k^T, \quad & \mathbf{M}_Y(t) = \frac{1}{\tau}\sum_{k=1}^{t-1}(\gamma^2)^{t-1-k}\mathbf{y}_k\mathbf{y}_k^T.
\end{aligned} \tag{8}
$$

## 5.2 Description of network dynamics for bounded anti-sparse sources

We now show that the gradient-descent minimization of the online WSM cost function in (6) can be interpreted as the dynamics of a neural network with local learning rules. The exact network architecture is determined by the presumed identifiable source domain $\mathcal{P}$, which can be chosen in infinitely many ways. In this section, we concentrate on the domain choice $\mathcal{P} = \mathcal{B}_\infty$ as an illustrative example. In Section 5.3, we discuss how to generalize the results of this section by modifying the output layer for different identifiable source domains. We start by writing the update expressions for the optimization variables based on the gradients of $\mathcal{J}(\mathbf{h}_t, \mathbf{y}_t, \mathbf{D}_1(t), \mathbf{D}_2(t))$:

Update dynamics for $\mathbf{h}_t$: Following previous work [48, 56], and using the gradient of (7) in (A.8) with respect to $\mathbf{h}_t$, we can write down an update dynamics for $\mathbf{h}_t$ in the form

$$\frac{d\mathbf{v}(\tau)}{d\tau} = -\mathbf{v}(\tau) - \lambda_{SM}[((1-\beta)\bar{\mathbf{M}}_H(t) + \beta\mathbf{D}_1(t)\bar{\mathbf{M}}_H(t)\mathbf{D}_1(t))\mathbf{h}(\tau)$$
$$+ \beta\mathbf{D}_1(t)\mathbf{W}_{HX}(t)\mathbf{x}(\tau) + (1-\beta)\mathbf{W}_{YH}(t)^T\mathbf{D}_2(t)\mathbf{y}(\tau)] \tag{9}$$

$$\mathbf{h}_{t,i}(\tau) = \sigma_A\left(\frac{\mathbf{v}_i(\tau)}{\lambda_{SM}\Gamma_{Hii}(t)((1-\beta)+\beta D_{1,ii}(t)^2)}\right), \quad \text{for } i = 1,\ldots n, \tag{10}$$

where $\mathbf{\Gamma}_H(t)$ is a diagonal matrix containing diagonal elements of $\mathbf{M}_H(t)$ and $\bar{\mathbf{M}}_H(t) = \mathbf{M}_H(t) - \mathbf{\Gamma}_H(t)$, $\sigma(\cdot)$ is the clipping function, defined as $\sigma_A(x) = \begin{cases} x & -A \le x \le A, \\ A\text{sign}(x) & \text{otherwise.} \end{cases}$ . This dynamics can be shown to minimize (7) [56]. Here $\mathbf{v}(\tau)$ is an internal variable that could be interpreted as the voltage dynamics of a biological neuron, and is defined based on a linear transformation of $\mathbf{h}_t$ in (A.9). Equation (9) defines $\mathbf{v}(\tau)$ dynamics from the gradient of (7) with respect to $\mathbf{h}_t$ in (A.8). Due to the positive definite linear map in (A.9), the expression in (A.8) also serves as a descent direction for $\mathbf{v}(\tau)$. Furthermore, $\sigma(\cdot)$ function is the projection onto $A\mathcal{B}_\infty$, where $[-A, A]$ is the presumed dynamic range for the components of $\mathbf{h}_t$. We note that there is no explicit constraint set for $\mathbf{h}_t$ in the online optimization setting of Section 5.1, and therefore, $A$ can be chosen as large as desired in the actual implementation. We included the nonlinearity in (10) to model the limited dynamic range of an actual (biological) neuron.

Update dynamics for output $\mathbf{y}_t$: We write the update dynamics for the output $\mathbf{y}_t$, based on (A.11) as

$$\frac{d\mathbf{u}(\tau)}{d\tau} = -\mathbf{u}(\tau) + \mathbf{W}_{YH}(t)\mathbf{h}(\tau) - \bar{\mathbf{M}}_Y(t)\mathbf{D}_2(t)\mathbf{y}(\tau), \tag{11}$$

$$\mathbf{y}_{t,i}(\tau) = \sigma_1\left(\frac{\mathbf{u}_i(\tau)}{\Gamma_{Yii}(t)D_{2,ii}(t)}\right), \quad \text{for } i = 1,\ldots n, \tag{12}$$

which is derived using the same approach for $\mathbf{h}_t$, where we used the descent direction expression in (A.11), and the substitution in (A.12). Here, $\mathbf{\Gamma}_Y(t)$ is a diagonal matrix containing diagonal elements of $\mathbf{M}_Y(t)$ and $\bar{\mathbf{M}}_Y(t) = \mathbf{M}_Y(t) - \mathbf{\Gamma}_Y(t)$. Note that the nonlinear mapping $\sigma_1(\cdot)$ is the projection onto the presumed domain of sources, i.e., $\mathcal{P} = \mathcal{B}_\infty$, which is elementwise clipping operation.

The state space representations in (9)-(10) and (11)-(12) correspond to a two-layer recurrent neural network with input $\mathbf{x}_t$, hidden layer activation $\mathbf{h}_t$, output layer activation $\mathbf{y}_t$, $\mathbf{W}_{HX}$ ($\mathbf{W}_{HX}^T$) and $\mathbf{W}_{YH}$ ($\mathbf{W}_{YH}^T$) are the feedforward (feedback) synaptic weight matrices for the first and the second layers, respectively, and $\bar{\mathbf{M}}_H$ and $\bar{\mathbf{M}}_Y$ are recurrent synaptic weight matrices for the first and the second layers, respectively. The corresponding neural network schematic is provided in Figure 2.(b). The gain and synaptic weight dynamics below describe the learning mechanism for this network:

Update dynamics for gains $D_{l,ii}$: Using the derivative of the cost function with respect to $D_{1,ii}$ in (A.13), we can write the dynamics corresponding to the gain variable $D_{1,ii}$ as

$$\mu_{D_1}\frac{dD_{1,ii}(t)}{dt} = -(\lambda_{SM}\beta)(\|\mathbf{M}_{Hi,:}\|_{\mathbf{D}_1(t)}^2 - \|\mathbf{W}_{HXi,:}\|_2^2) - (1-\lambda_{SM})\frac{1}{D_{1,ii}(t)}, \tag{13}$$

where $\mu_{D_1}$ corresponds to the learning time-constant. Similarly, for the gain variable $D_{2,ii}$, the corresponding coefficient dynamics expression based on (A.14) is given by

$$\mu_{D_2}\frac{dD_{2,ii}(t)}{dt} = -(\lambda_{SM}\beta)(\|\mathbf{M}_{Yi,:}\|_{\mathbf{D}_2(t)}^2 - \|\mathbf{W}_{YHi,:}\|_2^2) - (1-\lambda_{SM})\frac{1}{D_{2,ii}(t)}, \tag{14}$$

where $\mu_{D_2}$ corresponds to the learning time-constant.

The inverses of the inner product weights $D_{l,ii}$ correspond to homeostatic gain parameters. The inspection of the gain updates in (13) and (14) leads to an interesting observation: whether the corresponding gain is going to increase or decrease depends on the balance between the norms of the recurrent and the feedforward synaptic strengths, which are the statistical indicators of the recent output and input activations, respectively. Hence, the homeostatic gain of the neuron will increase (decrease) if the level of recent output activations falls behind (surpasses) the level of recent input

activations to balance input/output energy levels. The resulting dynamics align with the experimental homeostatic balance observed in biological neurons [57].

Based on the definitions of the synaptic weight matrices in (8), we can write their updates as

$$\mathbf{M}_H(t+1) = \gamma^2 \mathbf{M}_H(t) + (1-\gamma^2)\mathbf{h}_t\mathbf{h}_t^T, \quad \mathbf{M}_Y(t+1) = \gamma^2\mathbf{M}_Y(t) + (1-\gamma^2)\mathbf{y}_t\mathbf{y}_t^T, \quad (15)$$
$$\mathbf{W}_{HX}(t+1) = \gamma^2\mathbf{W}_{HX}(t) + (1-\gamma^2)\mathbf{h}_t\mathbf{x}_t^T, \quad \mathbf{W}_{YH}(t+1) = \gamma^2\mathbf{W}_{YH}(t) + (1-\gamma^2)\mathbf{y}_t\mathbf{h}_t^T.$$

These updates are local in the sense that they only depend on variables available to the synapse, and hence are biologically plausible.

### 5.3 Det-max WSM neural network examples for more general source domains

Det-Max Neural Network obtained for the source domain $\mathcal{P} = \mathcal{B}_\infty$ in Section 5.2 can be extended to more general identifiable source domains by only changing the output dynamics. In Appendix D, we provide illustrative examples for different identifiable domain choices. Table 1 summarizes the output dynamics obtained for the identifiable source domain examples in Figure 1.

Table 1: Example source domains from Figure 1 and the corresponding output dynamics.

| | Source Domain | Output Dynamics | Output Activation |
|---|---|---|---|
|  | $\mathcal{P} = \mathcal{B}_{\infty,+}$ Nonnegative Anti-sparse | $\mathbf{y}_{t,i}(\tau) = \sigma_+ \left( \frac{\mathbf{u}_i(\tau)}{\Gamma_{Yii}(t)D_{2,ii}(t)} \right)$ |  |
|  | $\mathcal{P} = \mathcal{B}_1$ Sparse | $\mathbf{y}_{t,i}(\tau) = \mathrm{ST}_{\lambda_1(\tau)} \left( \frac{u_i(\tau)}{\lambda_{SM}(1-\beta)\Gamma_{Yii}(t)D_{2,ii}(t)} \right)$ $\frac{da(\tau)}{d\tau} = -a(\tau) + \sum_{k=0}^n |\mathbf{y}_{t,k}(\tau)| - 1 + \lambda_1(\tau),$ $\lambda_1(\tau) = \mathrm{ReLU}(a(\tau))$ |  |
|  | $\mathcal{P} = \mathcal{B}_{1,+}$ Nonnegative Sparse | $\mathbf{y}_{t,i}(\tau) = \mathrm{ReLU} \left( \frac{u_i(\tau)}{\lambda_{SM}(1-\beta)\Gamma_{Yii}(t)D_{2,ii}(t)} - \lambda_1(\tau) \right)$ $\frac{da(\tau)}{d\tau} = -a(\tau) + \sum_{k=0}^n \mathbf{y}_{t,k}(\tau) - 1 + \lambda_1(\tau),$ $\lambda_1(\tau) = \mathrm{ReLU}(a(\tau))$ |  |
|  | $\mathcal{P} = \Delta$ Unit Simplex | $\mathbf{y}_{t,i}(\tau) = \mathrm{ReLU} \left( \frac{u_i(\tau)}{\lambda_{SM}(1-\beta)\Gamma_{Yii}(t)D_{2,ii}(t)} - \lambda_1(\tau) \right)$ $\frac{d\lambda_1(\tau)}{d\tau} = -\lambda_1(\tau) + \sum_{k=0}^n \mathbf{y}_{t,k}(\tau) - 1 + \lambda_1(\tau)$ |  |

We can make the following observations on Table 1: (1) For sparse and unit simplex settings, there is an additional inhibitory neuron which takes input from all outputs and whose activation is the inhibitory signal $\lambda_1(\tau)$, (2) The source attributes, which are globally defined over all sources, determine the activation functions at the output layer. The proposed framework can be applied to any polytope described by (1) for which the corresponding Det-Max neural network will contain combinations of activation functions in Table 1 as illustrated in Figure 2.(a).

## 6 Numerical experiments

In this section, we illustrate the applications of the proposed WSM-based BSS framework for both synthetic and natural sources. More details on these experiments and additional examples are provided in Appendix E, including sparse dictionary learning. Our implementation code is publicly available[1].

### 6.1 Synthetically correlated source separation

In order to illustrate the correlated source separation capability of the proposed WSM neural networks, we consider a numerical experiment with five copula-T distributed (uniform and correlated) sources. For the correlation calibration matrix for these sources, we use Toeplitz matrix whose first row is $[1 \quad \rho \quad \rho \quad \rho \quad \rho]$. The $\rho$ parameter determines the correlation level, and we considered the range $[0, 0.6]$ for this parameter. These sources are mixed with a $10 \times 5$ random matrix with independent and identically distributed (i.i.d.) standard normal random variables. The mixtures are corrupted by i.i.d.

---

[1]https://github.com/BariscanBozkurt/Biologically-Plausible-DetMaxNNs-for-Blind-Source-Separation

normal noise corresponding to 30dB signal-to-noise ratio (SNR) level. In this experiment, we employ the nonnegative-antisparse-WSM neural network (Figure 8 in Appendix D.2) whose activation functions at the output layer are nonnegative-clipping functions, as the sources are nonnegative uniform random variables.

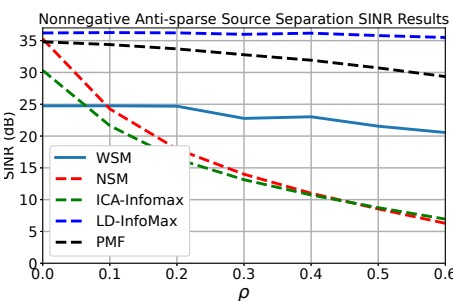

Figure 3: The SINRs of WSM, NSM, ICA, PMF, and LD-InfoMax versus the correlation parameter $\rho$.

We compared the signal-to-interference-plus-noise-power-ratio (SINR) performance of our algorithm with the NSM algorithm [16], Infomax ICA algorithm [1], as implemented in Python MNE Toolbox [58], LD-InfoMax algorithm [59], and PMF algorithm [26]. Figure 3 shows the SINR performances of these algorithms (averaged over 300 realizations) as a function of the correlation parameter $\rho$. We observe that our WSM-based network performs well despite correlations. In contrast, performance of NSM and ICA algorithms, which assume uncorrelated sources, degrade noticeably with increasing correlation levels. In addition, we note that the performance of batch Det-Max algorithms, i.e., LD-InfoMax and PMF, are also robust against source correlations. Furthermore, due to their batch nature, these algorithms typically achieved better performance results than our neural network with online-restriction, as expected.

## 6.2 Image separation

To further illustrate the correlated source separation advantage of our approach, we consider a natural image separation scenario. For this example, we have three RGB images with sizes $324 \times 432 \times 3$ as sources (Figure 4.(a)). The sample Pearson correlation coefficients between the images are $\rho_{12} = 0.263$, $\rho_{13} = 0.066$, $\rho_{23} = 0.333$. We use a random $5 \times 3$ mixing matrix whose entries are drawn from i.i.d standard normal distribution. The corresponding mixtures are shown in Figure 4.(b).

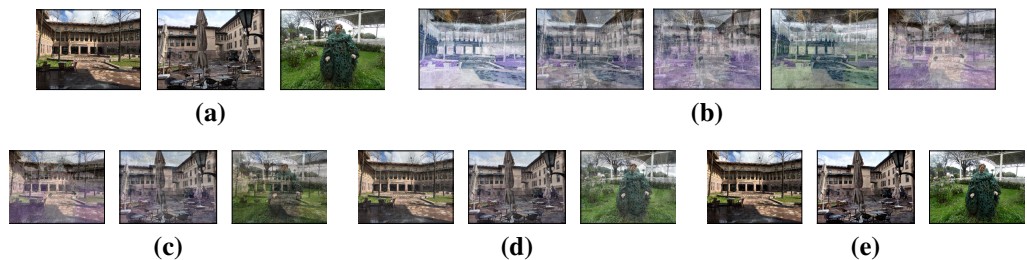

Figure 4: **(a)** Original RGB images, **(b)** Mixture RGB images, **(c)** ICA outputs, **(d)** NSM outputs (using pre-whitened mixtures), **(e)** WSM outputs.

We applied ICA, NSM and WSM algorithms to the mixtures. Figure 4.(c),(d),(e) shows the corresponding outputs. High-resolution versions of all images in this example are available in Appendix E.4 in addition to the comparisons with LD-Infomax and PMF algorithms. The Infomax ICA algorithm's outputs have SINR level of 13.92dB, and this performance is perceivable as residual interference effects in the corresponding output images. The NSM algorithm achieves significantly higher SINR level of 17.45dB and the output images visually reflect this better performance. Our algorithm achieves the best SINR level of 27.49dB, and the corresponding outputs closely resemble the original source images.

# 7 Discussion and Conclusion

We proposed a general framework for generating biologically plausible neural networks that are capable of separating correlated sources from their linear mixtures, and demonstrated their successful correlated source separation capability through synthetic and natural sources.

Another motivation for our work is to link network structure with function. This is a long standing goal of neuroscience, however examples where this link can be achieved are limited. Our work provides concrete examples where clear links between a network's architecture–i.e. number of interneurons, connections between interneurons and output neurons, nonlinearities (frequency-current curves)– and its function, the type of source separation or feature extraction problem the networks solves, can be established. These links may provide insights and interpretations that might generalize to real biological circuits.

Our networks suffer from the same limitations of other recurrent biologically-plausible BSS networks. First, certain hyperparameters can significantly influence algorithm performance (see Appendix E.9). Especially, the inner product gains ($D_{ii}$) are sensitive to the combined choices of algorithm parameters, which require careful tuning. Second, the numerical experiments with our neural networks are relatively slow due to the recursive computations in (9)-(10) and (11)-(12) for hidden layer and output vectors, which is common to all biologically plausible recurrent source separation networks (see Appendix F). This could perhaps be addressed by early-stopping the recursive computation [60].

## Acknowledgments and Disclosure of Funding

This work/research was supported by KUIS AI Center Research Award. C. Pehlevan acknowledges support from the Intel Corporation.

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
