# OpenReview forum: "Biologically-Plausible Determinant Maximization Neural Networks for Blind Separation of Correlated Sources"
_NeurIPS.cc/2022/Conference — NeurIPS 2022 Accept_

### Official Review · Reviewer_FU8C · 2022-06-30

**Rating:** 6
**Confidence:** 1
**Soundness:** 2 fair
**Presentation:** 1 poor
**Contribution:** 2 fair

**Summary:**

The paper proposes a novel neural network architecture to perform a blind source separation task, specifically addressing the case of correlated sources. The proposed architecture is shallow (2 or 3 layers) and derived in an online manner to maximize biologically plausibility. The proposed architecture is shown to solve a determinant-maximization problem, as proved in Theorem 1 (line 172), and to adequately solve two synthetically correlated source separation toy examples.

**Questions:**

1. What makes the proposed method well suited from correlated sources?

1. Figure 3/4 only compare the proposed method with ICA and NSM, despite a larger number of related works listed in section 1.1: NMF, SSMF, SCA, BCA, PMF, BSM. Why not comparing with these other methods?


**Limitations:**

- The paper stresses the computational cost of the proposed method, but does not give concrete examples. What is the computational cost of the proposed method (including hyperparameter tuning) in the two proposed numerical experiments? What is the computational cost of ICA and NSM?

**Strengths And Weaknesses:**

*Disclaimer: Due to my lack of expertise on the topic, I am not able to assess the soundness of the mathematical model nor the correctness of the theorem proof. I have also not read carefully the very long appendix (31 pages, 17 figures). Due to the length of the full manuscript, a journal might be a better publishing venue to benefit from more extensive reviews.*

### Originality
The blind source separation problem tackled by the paper is a well-known problem, addressed by a wide literature of methods. The originality seems to resides in the special case of correlated sources (as opposed to e.g. ICA which specifically assumes independent sources).

The proposed approach builds upon a recent framework called weighted similarity matching (WSM) and introduced in [15]. The difference with related works is very briefly discussed lines 70-74.

### Clarity
The paper is clearly written, but the relation to prior work is somewhat limited. A number of existing methods are listed, but the difference with the proposed method is not clearly explained. The paper would benefit from being more pedagogical about its different original contributions.

### Quality
The numerical experiments to demonstrate the effectiveness of the model are quite limited. It would have been interesting to compare the proposed method with more methods.

---

> ### Author Response · Authors · 2022-08-02
> **Response to Reviewer FU8C - Part 3 of 3**
>
>
> >Question 1: What makes the proposed method well suited from correlated sources?
>
> Correlated source separation capability is enabled by two factors: Using information about special source domain (such as unit simplex or identifiable polytope membership) and employing Det-Max criterion which assumes and exploits spreading of source samples inside the presumed domain. The use of these two factors is sufficient for source separation, eliminating the need for statistical assumptions such as the independence or uncorrelatedness of sources (see the end of Section 2.1, Section 3 and Appendix A for details).
>
> >Limitations: The paper stresses the computational cost of the proposed method, but does not give concrete examples. What is the computational cost of the proposed method (including hyperparameter tuning) in the two proposed numerical experiments? What is the computational cost of ICA and NSM?
>
> Thank you for incentivizing us to clarify our comment on the computational complexity. To address this issue, we included a section in the supplementary (Appendix F) on the computational complexity of the proposed Det-Max WSM neural networks. In this section, we derive the dimension (input, output) dependence for  the  per sample computational complexity of simulating these neural networks, which is given by $\mathcal{O}(\tau_{\text{max}} mn)$. Here, $n$ is the number of sources, $m$ is the number of mixtures and $\tau_{\text{max}}$ is the number of iterations required to obtain each output vector. We make comparisons with NSM and BSM, which are also biologically plausible neural networks for the BSS problem. We find that, because of their similar recurrent dynamics, their computational complexity are nearly the same except for constant scalings.  To summarize, the recurrent neural networks have  very similar dimension dependence. However, our proposed networks have two coupled layers  which results in the increased scaling factor.
>
> We note that the aforementioned complexity issue concerns only the digital computer based simulation of these neural networks. The major source of the complexity is on obtaining iterative numerical solutions of the differential equations in (9)-(12). In the analog/neuromorphic implementations, these differential equations are naturally solved by the physical circuit, without any computational load.

---

> ### Author Response · Authors · 2022-08-02
> **Response to Reviewer FU8C - Part 2 of 3**
>
> >Quality: The numerical experiments to demonstrate the effectiveness of the model are quite limited. It would have been interesting to compare the proposed method with more methods.
>
> >Question 2: Figure 3/4 only compare the proposed method with ICA and NSM, despite a larger number of related works listed in section 1.1: NMF, SSMF, SCA, BCA, PMF, BSM. Why not comparing with these other methods? Did you try any experiments with real data ?
>
> Following your suggestion (and the suggestions of other reviewers), in the revised version of our manuscript, we included comparisons with Polytopic Matrix Factorization (PMF) [25] and Log-Det Mutual Information Maximization (LD-InfoMax) [58] for nonnegative antisparse source separation (Section 6.1), antisparse source separation (Section E.2), image separation (Section E.3), and sparse source separation (Section E.4) experiments. Both PMF and LD-InfoMax use Det-Max criterion in Problem (3) to solve blind source separation problem for potentially correlated sources.
> These frameworks consider the off-line version of the  separation scenario we discussed in Sections 2.2 and 2.3, and propose batch algorithms for its solution.
>
>
> Polytopic Matrix Factorization is based on the following optimization problem:
> $$\text{maximize } \det(\boldsymbol{Y}(t) \boldsymbol{Y}(t)^T) \text{ subject to } \boldsymbol{X}(t) = \boldsymbol{H Y}(t), \text{ and } \boldsymbol{Y}(t)_{:,j} \in \mathcal{P} \ \ \forall j \ \ (R1),$$
> where $\boldsymbol{H}$ corresponds to the mixing matrix, $\boldsymbol{Y}(t)$ corresponds to the source estimates.
>
>
> LD-InfoMax [58] can be considered as a statistical interpretation of PMF approach, and it has the following optimization setting:
> $$\text{maximize } \frac{1}{2}\log\det(\boldsymbol{\hat{R}\_y} + \epsilon \boldsymbol{I} ) - \frac{1}{2}\log\det(\boldsymbol{\hat{R}\_y} - \boldsymbol{\hat{R}\_{yx}}(\epsilon \boldsymbol{I} + \boldsymbol{\hat{R}\_{x}})^{-1} \boldsymbol{\hat{R}\_{yx}}^T+ \epsilon \boldsymbol{I} ) \text{ subject to }  \boldsymbol{Y}(t)_{:,j} \in \mathcal{P} \ \ \forall j \ \ (R2),$$
> where $\boldsymbol{\hat{R}\_{y}}$ and $\boldsymbol{\hat{R}\_{yx}}$ correspond to the sample covariance matrix of the source estimates $\boldsymbol{Y}(t)$ and the sample cross-covariance matrix between source estimates and mixtures, respectively.
>
> We note that both the PMF algorithm in [25] and the LD-InfoMax algorithm in [58] are batch / off-line algorithms. In other words, at each iteration, these algorithms have access to all input data. Due to their batch nature, these algorithms typically achieved better performance results than our neural networks with online-restriction, as expected. We included the results of these new experiments in [Figure 3](https://figshare.com/s/07ebb57d19b77008e12c) (nonnegative antisparse case), [Figure 13](https://figshare.com/s/489eeb186c3d302630b1) (antisparse case) and Table 2 (sparse case).  Moreover, we added Section E.1 in our revised manuscript for a brief discussion of these batch algorithms.
>
> Regarding the use of real data:  In our article, we performed two experiments with natural images. The experiment in Section 6.2 uses real natural images that are mixed using a random mixing matrix. These images constitute nice examples of naturally correlated sources. Furthermore, the sparse dictionary learning example in Appendix E.6 is also based on the patches obtained from real images. Furthermore, this example forms a nice example for the computational modeling of the primary visual cortex through the proposed biologically plausible neural network based on the $\ell_1$-norm-ball polytope. Also, as an additional example, we provided a digital communication scenario in Section E.8 in our revised manuscript. Sources in digital communication are members of discrete sets, referred to as constellations, making the use of $\ell_\infty$-norm-ball polytope a perfect modeling assumption.  Furthermore, random mixing matrices with identically distributed independent normal entries are common, and in many cases accurate models of the actual wireless propagation environment [i].  Therefore,  simulations involving a wireless digital communication scenario using discrete sources and random matrix mixing would be highly realistic. We presented the simulation results and demonstrate that the proposed neural network successfully handles multi-user separation task at a
> receiver.
>
> [i] Jakes WC, Cox DC, editors. Microwave mobile communications. Wiley-IEEE press; 1994 Sep 1.

---

> ### Author Response · Authors · 2022-08-02
> **Response to Reviewer FU8C - Part 1 of 3**
>
> We thank you for your time and your comments. Below we present our response to your comments and questions.
>
> >Disclaimer by Reviewer: Due to my lack of expertise on the topic, I am not able to assess the soundness of the mathematical model nor the correctness of the theorem proof. I have also not read carefully the very long appendix (31 pages, 17 figures). Due to the length of the full manuscript, a journal might be a better publishing venue to benefit from more extensive reviews.
>
> We appreciate your recommendation. We respectfully think that our article is a good fit for Neurips, which has published many articles in recent years on the topic of biologically-plausible neural networks [i,ii,iii], and many papers with similar or longer lengths.
>
> [i]. Bahroun Y, Chklovskii D, Sengupta A. A Normative and Biologically Plausible Algorithm for Independent Component Analysis. Advances in Neural Information Processing Systems. 2021 Dec 6;34:7368-84.
>
> [ii]. Pogodin R, Mehta Y, Lillicrap T, Latham PE. Towards biologically plausible convolutional networks. Advances in Neural Information Processing Systems. 2021 Dec 6;34:13924-36.
>
> [iii]. Tyulmankov D, Fang C, Vadaparty A, Yang GR. Biological learning in key-value memory networks. Advances in Neural Information Processing Systems. 2021 Dec 6;34:22247-58.
>
> >Clarity: The paper is clearly written, but the relation to prior work is somewhat limited. A number of existing methods are listed, but the difference with the proposed method is not clearly explained. The paper would benefit from being more pedagogical about its different original contributions.
>
> We thank you for the positive feedback on the presentation of our article. Section 1.1 (Other related work) of our paper provides background on relevant existing methods and describes contributions of our article in relation to them. Following your suggestion, in the revised article, before Section 1.1, we added a list of the main contributions of the article:
>
>  "In summary, our main contributions in this article are the following:
>
> * We propose a normative framework for generating biologically plausible neural networks that are capable of separating correlated sources from their mixtures by deriving them from a Det-Max objective function subject to source domain constraints.
> * Our framework can handle infinitely many source types by exploiting their source domain topology.
> * We demonstrate the performance of our networks in simulations with synthetic and realistic data.
> "
>
>
> To provide more clarification,
>
> Main novelty : Our article proposes a general framework for constructing biologically plausible neural networks that are capable of separating correlated sources based on information about their source domains. This framework utilizes the Det-Max criterion for correlated source separation capability and two-layer weighted similarity matching to construct biologically plausible neural networks capable of implementing arbitrary linear transformations.
>
> Comparison with existing approaches:
>
> * There are BSS frameworks that are capable of separating correlated sources using the Det-Max criterion and source domain information such as  SSMF, PMF and LD-InfoMax. However, the algorithms for these frameworks are not online, but batch algorithms.  In fact, the proposed framework enables biologically plausible neural network-based (online) implementation of these frameworks.
> * There are biologically plausible neural network-based  BSS solvers derived using the similarity matching criterion as Nonnegative Similarity Matching (NSM) and Bounded Similarity Matching (BSM). However, these approaches cannot handle correlated sources. BSM employs a single-layer WSM, which is unable to implement arbitrary linear transformations. Furthermore, they are specific to only two source domains: nonnegative orthant for NSM and hypercube ($\ell_\infty$-norm ball) for BSM. The proposed framework is capable of handling infinitely many source domains including the unit simplex and infinitely many polytopes corresponding to different source domain characteristics.

---

> ### Comment · Reviewer_FU8C · 2022-08-08
> **Answer to authors**
>
> The authors have adequately answered my comments, I have raised my rating accordingly.

---

> > ### Author Response · Authors · 2022-08-09
> > **Thank you**
> >
> > We would like to thank you for your useful feedback and suggestions.

---

### Official Review · Reviewer_NHdM · 2022-07-06

**Rating:** 8
**Confidence:** 5
**Soundness:** 4 excellent
**Presentation:** 3 good
**Contribution:** 4 excellent

**Summary:**

This work follows a recent line of work on formulating blind source separation problems as solutions of similarity matching objective functions. This work greatly expands existing work in the domain by proposing geometric interpretation and an objective function related to the Det-Max approach. Also, the formalism allows for the derivation of a biologically-plausible and online learning algorithm. Indeed, the model can be implemented by a two-layer neural network with local learning rules.

**Questions:**

My questions relate to the minor weaknesses found above:

Can you propose a concise presentation of the online result showing that your model can operate in that setting?

Can you explain why would biological plausible model has to solve such problems?

Can you compare your model to existing algorithms that were designed to solve it, and not only bio-inspired ones?

**Limitations:**

The authors have addressed the limitations of the paper in the last section.

**Strengths And Weaknesses:**

The authors have covered a broad class of blind source separation problems. These problems generalize the well-known problems for which there existed or not biologically plausible learning algorithms. There is a broad modeling literature on BSS, and a growing one on similarity matching, and these generalizations and extensions of both is very novel. Although it is very technical the manuscript is well written and is easy fairly easy to follow.

I found three minor weaknesses in this paper that can be easily addressed.

One is that the paper claims that the resulting algorithm is only, but only presents these results in the appendix. I would have appreciated that some space of the main paper be allocated to that as it is rather central to the paper.

The second one is related to the context of the problem. The paper is at the interface of signal processing, machine learning, and neuroscience, and it is a bit much to ask the reader to be well versed in all the different BSS problems covered in the paper. A bit more context for each of the problems would help understand the importance of each of these problems and why building such biologically plausible would be useful. Are natural data mixed or present in the form presented in this paper?

Finally, the work mainly compares to nonnegative similarity matching and infomax, but it could be interesting to see how the model compares to existing algorithms designed to solve the problem at hand, not only biologically inspired ones.

---

> ### Author Response · Authors · 2022-08-02
> **Response to Reviewer NHdM - Part 3 of 3**
>
>
> > Weaknesses 3: Finally, the work mainly compares to nonnegative similarity matching and infomax, but it could be interesting to see how the model compares to existing algorithms designed to solve the problem at hand, not only biologically inspired ones.
>
> > Question 3: Can you compare your model to existing algorithms that were designed to solve it, and not only bio-inspired ones?
>
> Following your suggestion (and the suggestions of other reviewers), in the revised version of our manuscript, we included comparisons with Polytopic Matrix Factorization (PMF) [25] and Log-Det Mutual Information Maximization (LD-InfoMax) [58] for nonnegative antisparse source separation (Section 6.1), antisparse source separation (Section E.2), image separation (Section E.3), and sparse source separation (Section E.4) experiments. Both PMF and LD-InfoMax use Det-Max criterion in Problem (3) to solve blind source separation problem for potentially correlated sources.
> These frameworks consider the off-line version of the  separation scenario we discussed in Sections 2.2 and 2.3, and propose batch algorithms for its solution.
>
>
> Polytopic Matrix Factorization is based on the following optimization problem:
> $$\text{maximize } \det(\boldsymbol{Y}(t) \boldsymbol{Y}(t)^T) \text{ subject to } \boldsymbol{X}(t) = \boldsymbol{H Y}(t), \text{ and } \boldsymbol{Y}(t)_{:,j} \in \mathcal{P} \ \ \forall j \ \ (R1),$$
> where $\boldsymbol{H}$ corresponds to the mixing matrix, $\boldsymbol{Y}(t)$ corresponds to the source estimates.
>
>
> LD-InfoMax [58] can be considered as a statistical interpretation of PMF approach, and it has the following optimization setting:
> $$\text{maximize } \frac{1}{2}\log\det(\boldsymbol{\hat{R}\_y} + \epsilon \boldsymbol{I} ) - \frac{1}{2}\log\det(\boldsymbol{\hat{R}\_y} - \boldsymbol{\hat{R}\_{yx}}(\epsilon \boldsymbol{I} + \boldsymbol{\hat{R}\_{x}})^{-1} \boldsymbol{\hat{R}\_{yx}}^T+ \epsilon \boldsymbol{I} ) \text{ subject to }  \boldsymbol{Y}(t)_{:,j} \in \mathcal{P} \ \ \forall j \ \ (R2),$$
> where $\boldsymbol{\hat{R}\_{y}}$ and $\boldsymbol{\hat{R}\_{yx}}$ correspond to the sample covariance matrix of the source estimates $\boldsymbol{Y}(t)$ and the sample cross-covariance matrix between source estimates and mixtures, respectively.
>
> We note that both the PMF algorithm in [25] and the LD-InfoMax algorithm in [58] are batch / off-line algorithms. In other words, at each iteration, these algorithms have access to all input data. Due to their batch nature, these algorithms typically achieved better performance results than our neural networks with online-restriction, as expected. We included the results of these new experiments in [Figure 3](https://figshare.com/s/07ebb57d19b77008e12c) (nonnegative antisparse case), [Figure 13](https://figshare.com/s/489eeb186c3d302630b1) (antisparse case) and Table 2 (sparse case).  Moreover, we added Section E.1 in our revised manuscript for a brief discussion of these batch algorithms.

---

> > ### Comment · Reviewer_NHdM · 2022-08-05
> > **Response to the authors**
> >
> > I have satisfied with the response and changes made by the authors. I keep my positive review.

---

> > > ### Author Response · Authors · 2022-08-09
> > > **Thank you**
> > >
> > > We would like to thank you for your useful feedback and suggestions.

---

> ### Author Response · Authors · 2022-08-02
> **Response to Reviewer NHdM - Part 2 of 3**
>
> >Weaknesses 2: The second one is related to the context of the problem. The paper is at the interface of signal processing, machine learning, and neuroscience, and it is a bit much to ask the reader to be well versed in all the different BSS problems covered in the paper. A bit more context for each of the problems would help understand the importance of each of these problems and why building such biologically plausible would be useful. Are natural data mixed or present in the form presented in this paper?
>
> We thank the reviewer for this question. The main motivation for this work comes from the fact that blind source separation may be implemented throughout the brain. For example, seminal work argued that Gabor receptive
>  fields of V1 neurons may be the result of performing BSS on natural images [1, 2], and receptive fields in the auditory system may be the result of performing BSS on natural sounds [4]. In addition, there may be general circuit motifs in the brain for solving BSS. For example, in a seminal experiment, auditory cortex neurons acquired V1-like receptive
> fields when visual inputs were redirected there in a ferret [14], suggesting that auditory and visual cortices may be implementing similar learning algorithms. Many previous works cited in our paper were motivated by these points and developed biologically plausible BSS algorithms.
>
> We modified to the first paragraph of our introduction to address these points:
> "Our brains constantly and effortlessly extract latent causes, or sources, of complex visual, auditory or olfactory stimuli sensed by sensory organs [1-11]. This extraction is mostly done without any instruction, in an unsupervised manner, making the process an instance of the blind source separation (BSS) problem [12-13]. Indeed, visual and auditory cortical receptive fields were argued to  be the result of performing BSS on natural images [1-2] and sounds [4]. The wide-spread use of BSS in the brain suggests the existence of generic circuit motifs that perform this task [14]. Consequently, the literature on biologically-plausible neural network algorithms for BSS is growing [15-19]."
>
> The main contribution of this paper is the consideration of correlated sources, which have not been addressed previously. To solve these correlated source separation problems, we make geometric assumptions about the sources. These assumptions naturally map to biological sources. For example, boundedness is a reasonable assumption for natural sources. Nonnegativity is also natural; in an olfactory mixture, odorants are either there or not. Sparseness is a feature of wavelet-like representations of natural scenes [2]. Antisparseness is motivated by dense or democratic representations [i],[ii], which might be suitable for some internal or sensory representations.
>
> We modified the sentence in line 87 to "The use of $\ell_1$-norm as a convex (non)sparsity measure has been quite successful with various applications including sparse dictionary learning/component analysis [29, 38, 40, 48, 49] and modeling of V1 receptive fields [2]."  and the sentence in line 97 to "Nonnegativity of sources naturally arises in biological context, for example in demixing olfactory mixtures [53]."
>
> [i] Studer C, Goldstein T, Yin W, Baraniuk RG. Democratic representations. arXiv preprint arXiv:1401.3420. 2014 Jan 15.
>
> [ii] Studer C, Yin W, Baraniuk RG. Signal representations with minimum ℓ∞-norm. In2012 50th Annual Allerton Conference on Communication, Control, and Computing (Allerton) 2012 Oct 1 (pp. 1270-1277). IEEE.
>
> Another motivation for our work is to link network structure with function. This is a long standing goal of neuroscience, however examples where this link can be achieved are limited. Our work provides concrete examples where clear links between a network's architecture--i.e. number of interneurons, connections between interneurons and output neurons, nonlinearities (frequency-current curves)-- and its function, the type of source separation or feature extraction problem the networks solves, can be established. These links may provide insights and interpretations that might generalize to real biological circuits.
>
> Unfortunately, space limitations do not allow us to further expand our discussion. If we get an extra page in the final submission, we will add more on this topic.

---

> ### Author Response · Authors · 2022-08-02
> **Response to Reviewer NHdM - Part 1 of 3**
>
>  We thank you for your time and for the general positive assessment of our article. We address your comments and questions in the revised article and in our response below.
>
> >Weaknesses 1: One is that the paper claims that the resulting algorithm is only, but only presents these results in the appendix. I would have appreciated that some space of the main paper be allocated to that as it is rather central to the paper.
>
> >Question: Can you propose a concise presentation of the online result showing that your model can operate in that setting?
>
> Thank you for this comment. We would like to clarify the coverage of the online aspect of our framework in the main part of the article:
>
> * We provide the generic online optimization setting in Section 5.1 of the article.
> * Only the gradient expressions corresponding to the objective function are provided in the Appendix.
> * We illustrate the derivation of the online algorithm for a special source case, i.e., antisparse sources represented by the $\ell_\infty$-norm-ball domain, in Section 5.2.
> * Online algorithm is defined as a gradient search, which is implemented in the form of the output dynamics in (9)-(12) and learning updates in (13)-(15). All of these equations correspond to the gradient search for the online optimization setting in Section 5.1.
> * All these components of the online algorithm define our neural network structure, which is illustrated in Figure 2.1.
>
>  In summary, we provide the derivation of a neural network determined by the gradient search-based online algorithm for the special source domain of anti-sparse sources in the main part of our article. Other example source domains are provided in the supplementary part.

---

### Official Review · Reviewer_ffJU · 2022-07-11

**Rating:** 7
**Confidence:** 3
**Soundness:** 3 good
**Presentation:** 3 good
**Contribution:** 3 good

**Summary:**

This work focuses on the BSS problem and solves it by imposing some geometrical priors on the sources via an online weighted similarity matching algorithm (WSM).
Since WSM does not use statistical independence of the sources to recover them, sources can be recovered even if they are correlated.
WSM is benchmarked on 2 datasets (a synthetic and a toy dataset) and is shown to yield better performance than ICA or non-negative similarity matching (NSM).

**Questions:**

- Hyper-parameters setting:
There are many hyper-parameters to set in this method.  Did you perform a sensitivity analysis to see in which range they work ? Is there a way to find default values that would work for all kind of problems ? Can you explain how you chose the values in section E.2 and E.3 ?

- Experiments on synthetic data:
While the experiments on synthetic data show what they are meant to show, it would have been nice to see some experiments with real systems. ICA is used in many different settings: astronomy, neuroscience, finance (see the section 7 of Hyvärinen, Aapo, and Erkki Oja. "Independent component analysis: algorithms and applications." Neural networks 13.4-5 (2000): 411-430.).
Did you try any experiments with real data ?

- Missing comparison with Det-Max criterion
According to your theorem 1: Problem (3) with Det-Max criterion and Problem (4) that yield the equations for WSM updates solve essentially the same problem. Therefore it would have seem natural to check whether they yield the same results in the synthetic experiments. Did you perform this comparison ?
The Det-Max criterion is mathematically much simpler and seem easier to optimize. Why should practitioners use WSM instead ?

**Limitations:**

Ethical limitations are properly discussed

**Strengths And Weaknesses:**

Strength:
- A general framework that applies to a large set of priors on the sources
- An online algorithm with local update rules that is therefore more biologically plausible

Weaknesses:
- Many hyper-parameters to set and no clear rules on how to set them (but the paper is transparent about this which is a good point)
- Experiments are on synthetic data and a toy dataset that does not really correspond to any realistic problem
- Missing comparison with Det-Max criterion

I quickly reviewed the code: it lacks documentation and unit tests are missing but is overall well structured and readable. I would still advice the authors to document every public function, make unit tests, examples and set up a continuous integration so that other researchers can easily build upon their work.

---

> ### Author Response · Authors · 2022-08-02
> **Response to Reviewer ffJU - Part 2 of 2**
>
> >Weaknesses 3: Missing comparison with Det-Max criterion.
>
> > Question 3: Missing comparison with Det-Max criterion According to your theorem 1: Problem (3) with Det-Max criterion and Problem (4) that yield the equations for WSM updates solve essentially the same problem. Therefore it would have seem natural to check whether they yield the same results in the synthetic experiments. Did you perform this comparison ? The Det-Max criterion is mathematically much simpler and seem easier to optimize. Why should practitioners use WSM instead ?
>
> Following your suggestion (and the suggestions of other reviewers), in the revised version of our manuscript, we included comparisons with Polytopic Matrix Factorization (PMF) [25] and Log-Det Mutual Information Maximization (LD-InfoMax) [58] for nonnegative antisparse source separation (Section 6.1), antisparse source separation (Section E.2), image separation (Section E.3), and sparse source separation (Section E.4) experiments. Both PMF and LD-InfoMax use Det-Max criterion in Problem (3) to solve blind source separation problem for potentially correlated sources.
> These frameworks consider the off-line version of the  separation scenario we discussed in Sections 2.2 and 2.3, and propose batch algorithms for its solution.
>
>
> Polytopic Matrix Factorization is based on the following optimization problem:
> $$\text{maximize } \det(\boldsymbol{Y}(t) \boldsymbol{Y}(t)^T) \text{ subject to } \boldsymbol{X}(t) = \boldsymbol{H Y}(t), \text{ and } \boldsymbol{Y}(t)_{:,j} \in \mathcal{P} \ \ \forall j \ \ (R1),$$
> where $\boldsymbol{H}$ corresponds to the mixing matrix, $\boldsymbol{Y}(t)$ corresponds to the source estimates.
>
>
> LD-InfoMax [58] can be considered as a statistical interpretation of PMF approach, and it has the following optimization setting:
> $$\text{maximize } \frac{1}{2}\log\det(\boldsymbol{\hat{R}\_y} + \epsilon \boldsymbol{I} ) - \frac{1}{2}\log\det(\boldsymbol{\hat{R}\_y} - \boldsymbol{\hat{R}\_{yx}}(\epsilon \boldsymbol{I} + \boldsymbol{\hat{R}\_{x}})^{-1} \boldsymbol{\hat{R}\_{yx}}^T+ \epsilon \boldsymbol{I} ) \text{ subject to }  \boldsymbol{Y}(t)_{:,j} \in \mathcal{P} \ \ \forall j \ \ (R2),$$
> where $\boldsymbol{\hat{R}\_{y}}$ and $\boldsymbol{\hat{R}\_{yx}}$ correspond to the sample covariance matrix of the source estimates $\boldsymbol{Y}(t)$ and the sample cross-covariance matrix between source estimates and mixtures, respectively.
>
> We note that both the PMF algorithm in [25] and the LD-InfoMax algorithm in [58] are batch / off-line algorithms. In other words, at each iteration, these algorithms have access to all input data. Due to their batch nature, these algorithms typically achieved better performance results than our neural networks with online-restriction, as expected. We included the results of these new experiments in [Figure 3](https://figshare.com/s/07ebb57d19b77008e12c) (nonnegative antisparse case), [Figure 13](https://figshare.com/s/489eeb186c3d302630b1) (antisparse case) and Table 2 (sparse case).  Moreover, we added Section E.1 in our revised manuscript for a brief discussion of these batch algorithms.
>
> Note that the reason for the introduction of the WSM setting in (4) in Section 4, instead of direct use of the PMF or LD-InfoMax batch algorithms to solve the Det-Max problem in (3), is to be able to produce online optimization formulation in (5) and (6), which leads to biologically plausible neural networks with local update rules. Furthermore, we note that, although the implicit definition of the algorithm's output (see Algorithm 1 in Section E) for such biologically plausible neural networks makes the implementation less practical in digital hardware, they enable efficient low-power implementations in future analog neuromorphic systems with local learning rule constraints.

---

> > ### Comment · Reviewer_ffJU · 2022-08-08
> > **Missing comparison**
> >
> > My comment has been addressed here. I upgraded my grade.

---

> > > ### Author Response · Authors · 2022-08-09
> > > **Thank you**
> > >
> > > We would like to thank you for the useful feedback and suggestions.  In the final version of our article, we will extend  hyperparameter sensitivity  analysis in the appendix section. As one of the near future extensions of our framework, we will consider applications involving the use of real data.

---

> ### Author Response · Authors · 2022-08-02
> **Response to Reviewer ffJU - Part 1 of 2**
>
> We thank you for your useful feedback and positive reviews. We respond to each of your questions and comments below.
>
> >Weaknesses 1: Many hyper-parameters to set and no clear rules on how to set them (but the paper is transparent about this which is a good point).
>
> >Question 1: Hyper-parameters setting: There are many hyper-parameters to set in this method. Did you perform a sensitivity analysis to see in which range they work ? Is there a way to find default values that would work for all kind of problems ? Can you explain how you chose the values in section E.2 and E.3 ?
>
> Thank you for this comment. We agree that our Det-Max WSM framework uses several hyperparameters. Based on your comments, we include a new Appendix E.9 in the revised article to study the impact of certain hyperparameters that we observed to be relatively sensitive.
>
> The Det-Max WSM networks have fairly reasonable performance around their nominal values that we chose for our experiments. To determine these nominal values, we first determined the initial values of hyper-parameters with trial and error. Then we performed greedy search by optimizing the parameters via changing one at a time. The best choice of hyper-parameters we obtained for our initial source domain also constituted as a good starting point for greedy hyper-parameter optimization in other source domains.
>
> In section Appendix E.9 in our revised manuscript, we provide two brief ablation studies for the selection of $\lambda_{\text{SM}}$ and initialization of $\boldsymbol{D}\_1$. [Figure 23a](https://figshare.com/s/e666f43ca5631b7eb280) illustrates that the performance of our approach is relatively sensitive to the selection of $\lambda_{\text{SM}}$, and we obtain a near-optimal value $\lambda_{\text{SM}} = 1 - 10^{-4}$ as a result of this parameter search experiment. [Figure 23b](https://figshare.com/s/cba59736bcb6f05f04d7) demonstrates that the our network performance is less sensitive to the initialization of $\boldsymbol{D}\_1$ compared to $\lambda_{\text{SM}}$ since the algorithm relatively maintains its performance for different choice of gain variables.
>
> > Weaknesses 2: Experiments are on synthetic data and a toy dataset that does not really correspond to any realistic problem.
>
> >Question 2: Experiments on synthetic data: While the experiments on synthetic data show what they are meant to show, it would have been nice to see some experiments with real systems. ICA is used in many different settings: astronomy, neuroscience, finance (see the section 7 of Hyvärinen, Aapo, and Erkki Oja. "Independent component analysis: algorithms and applications." Neural networks 13.4-5 (2000): 411-430.). Did you try any experiments with real data ?
>
> Thanks for your suggestion. In our article, we performed two experiments with natural images. The experiment in Section 6.2 uses real natural images that are mixed using a random mixing matrix. These images constitute examples of naturally correlated sources. Furthermore, the sparse dictionary learning example in Appendix E.6 is also based on the patches obtained from real images. Furthermore, this example forms an example for the computational modeling of the primary visual cortex through the proposed biologically plausible neural network based on the $\ell_1$-norm-ball polytope.
>
> Following your recommendation, we added an additional example from digital communication systems.  Sources in digital communication are members of discrete sets, referred to as constellations, making the use of $\ell_\infty$-norm-ball polytope a perfect modeling assumption.  Furthermore, random mixing matrices with identically distributed independent normal entries are common, and in many cases accurate models of the actual wireless propagation environment [i].  Therefore,  simulations involving a wireless digital communication scenario using discrete sources and random matrix mixing would be highly realistic. In Appendix E.8. of the revised article, we provide such a simulation and demonstrate that the proposed neural network successfully handles multi-user  separation task at a receiver. Note that the use of neural networks with local learning rules is still relevant for such tasks, as they enable low-power/low-complexity implementations in future neuromorphic systems with local learning constraints.
>
> [i] Jakes WC, Cox DC, editors. Microwave mobile communications. Wiley-IEEE press; 1994 Sep 1.

---

> > ### Comment · Reviewer_ffJU · 2022-08-08
> > **Hyper parameter selection and experiments on real data**
> >
> > Thanks for your answers.
> > 1. Hyper-parameter selection.
> > Thanks for this new experiment. It shows the sensitivity of the experiment to the parameter $\lambda_{SM}$ and shows that the Gain initialization seems to have a limited impact. However, a lot of hyper-parameters are still fixed and for instance the rule where the Euclidean norm of all rows of W is normalized to 0.0033 seems rather arbitrary. My take in general is that "trial and errors" with many hyper-parameters is not desirable because it makes the method much longer to train if a proper cross validation is performed and otherwise there is a risk of over-fitting the test set.
> > 2. Experiments on real data
> > I strongly disagree that mixing images with random mixing matrices with entries sampled from a Gaussian distribution is real data. In this experiment, your model perfectly holds so you don't test robustness with respect to model mis-specification. For instance, in real applications, you might have additional noise that has nothing to do with the sources you are trying to recover or mixing matrices could be generated from any distribution.
> > The same thing holds for your additional experiments where  again, the mixing is fully artificial. In a real data experiment, you do not have access to how the data are mixed (think of a recording of several voices in a room, there can be reverberations, time delays and so many other disturbance that makes the model hold only partially).

---

### Official Review · Reviewer_qu9X · 2022-07-11

**Rating:** 6
**Confidence:** 3
**Soundness:** 3 good
**Presentation:** 3 good
**Contribution:** 3 good

**Summary:**

The authors present a biologically-plausible algorithm for blind source separation (BSS) of correlated input sources. By applying weight similarity matching (WSM) approach to the Det-Max optimization algorithm used for BSS, the authors show that they can derive 2-layered Hebbian neural networks that are able to separate correlated sources from linear mixtures. They then compare their methods against Independent-Components Analysis (ICA) and Nonnegative Similarity Matching (NSM) methods in two tasks: one with artificial signals and another with natural images. In the task with artificial signals, the authors show how their algorithms is robust against correlation in the input sources whereas ICA and NSM degrade as measured by SINRs. The authors report results for the task of separating mixtures of natural images that are in-line with the former results, and they cherry-pick an example that illustrated the superior performance of their algorithm.



**Questions:**

Please add more experiments and report SNIR comparing against algorithms that are able to separate correlated sources.

**Limitations:**

Yes.

**Strengths And Weaknesses:**

The paper offers a general framework for deriving neural-network from the Det-Max approach, which is a novel contribution. They are the first (to the best of my knowledge) to propose a biologically-plausible algorithm that separates correlated sources. Overall, seems a worth contribution in terms of novelty. In terms of quality, I think the paper is very solid, and there are no obvious errors that I can see. The only experiment that I am missing is one that compares WSM with algorithms that are able to separate correlated sources and reporting how the performance compares there. I think this could be a necessary addition to judge rating of the paper. The paper is fairly clearly written, but I would suggest to add a section that states more clearly what the contributions of the paper are. I like that the authors highlight the limitations of their approach. In terms of significance, I think the paper is relevant for understanding how brains may accomplish BSS and to investigate novel biologically-inspired algorithmic improvements that can help advance state-of-the-art methods of BSS with correlated sources.

---

> ### Author Response · Authors · 2022-08-02
> **Response to Reviewer qu9X**
>
> We thank you for your time and for your detailed review. We appreciate your rating our framework and results as solid. In the revised article and the comments below, we address your comments and concerns.
>
> > Novelty : The paper offers a general framework for deriving neural-network from the Det-Max approach, which is a novel contribution. They are the first (to the best of my knowledge) to propose a biologically-plausible algorithm that separates correlated sources.
>
> We thank you for the positive feedback.
>
> > Experimental Evaluation: The only experiment that I am missing is one that compares WSM with algorithms that are able to separate correlated sources and reporting how the performance compares there. I think this could be a necessary addition to judge rating of the paper.
>
> > Questions: Please add more experiments and report SNIR comparing against algorithms that are able to separate correlated sources.
>
> Following your suggestion (and the suggestions of other reviewers), in the revised version of our manuscript, we included comparisons with Polytopic Matrix Factorization (PMF) [25] and Log-Det Mutual Information Maximization (LD-InfoMax) [58] for nonnegative antisparse source separation (Section 6.1), antisparse source separation (Section E.2), image separation (Section E.3), and sparse source separation (Section E.4) experiments. Both PMF and LD-InfoMax use Det-Max criterion in Problem (3) to solve blind source separation problem for potentially correlated sources.
> These frameworks consider the off-line version of the  separation scenario we discussed in Sections 2.2 and 2.3, and propose batch algorithms for its solution.
>
>
> Polytopic Matrix Factorization is based on the following optimization problem:
> $$\text{maximize } \det(\boldsymbol{Y}(t) \boldsymbol{Y}(t)^T) \text{ subject to } \boldsymbol{X}(t) = \boldsymbol{H Y}(t), \text{ and } \boldsymbol{Y}(t)_{:,j} \in \mathcal{P} \ \ \forall j \ \ (R1),$$
> where $\boldsymbol{H}$ corresponds to the mixing matrix, $\boldsymbol{Y}(t)$ corresponds to the source estimates.
>
>
> LD-InfoMax [58] can be considered as a statistical interpretation of PMF approach, and it has the following optimization setting:
> $$\text{maximize } \frac{1}{2}\log\det(\boldsymbol{\hat{R}\_y} + \epsilon \boldsymbol{I} ) - \frac{1}{2}\log\det(\boldsymbol{\hat{R}\_y} - \boldsymbol{\hat{R}\_{yx}}(\epsilon \boldsymbol{I} + \boldsymbol{\hat{R}\_{x}})^{-1} \boldsymbol{\hat{R}\_{yx}}^T+ \epsilon \boldsymbol{I} ) \text{ subject to }  \boldsymbol{Y}(t)_{:,j} \in \mathcal{P} \ \ \forall j \ \ (R2),$$
> where $\boldsymbol{\hat{R}\_{y}}$ and $\boldsymbol{\hat{R}\_{yx}}$ correspond to the sample covariance matrix of the source estimates $\boldsymbol{Y}(t)$ and the sample cross-covariance matrix between source estimates and mixtures, respectively.
>
> We note that both the PMF algorithm in [25] and the LD-InfoMax algorithm in [58] are batch / off-line algorithms. In other words, at each iteration, these algorithms have access to all input data. Due to their batch nature, these algorithms typically achieved better performance results than our neural networks with online-restriction, as expected. We included the results of these new experiments in [Figure 3](https://figshare.com/s/07ebb57d19b77008e12c) (nonnegative antisparse case), [Figure 13](https://figshare.com/s/489eeb186c3d302630b1) (antisparse case) and Table 2 (sparse case).  Moreover, we added Section E.1 in our revised manuscript for a brief discussion of these batch algorithms.
>
> >Contribution Section: The paper is fairly clearly written, but I would suggest to add a section that states more clearly what the contributions of the paper are.
>
> Thank you for your positive assessment. The second paragraph of Section 1.1 (Other related work) contains the contributions of our article in relation to the existing literature. However, based on your suggestion, we added a list of main contributions of the article before Section 1.1:
>
>  "In summary, our main contributions in this article are the following:
>
> * We propose a normative framework for generating biologically plausible neural networks that are capable of separating correlated sources from their mixtures by deriving them from a Det-Max objective function subject to source domain constraints.
> * Our framework can handle infinitely many source types by exploiting their source domain topology.
> * We demonstrate the performance of our networks in simulations with synthetic and realistic data.
> "

---

> > ### Comment · Reviewer_qu9X · 2022-08-08
> > **Response to Conference Paper2913 Authors**
> >
> > I want to thank the authors for addressing all my comments. I think that the added results are very interesting and highlight the online nature of the algorithm. I think they are a very valuable addition to the paper. Also appreciate that the efforts made an effort to improve the legibility of the paper more by stating the contributions more clearly. Given the author's response and the other reviews, I see no reason to change my ratings.

---

> > > ### Author Response · Authors · 2022-08-09
> > > **Thank you**
> > >
> > > We would like to thank you for your useful feedback and suggestions.

---

### Meta-Review · Area_Chair_ptrV · 2022-08-24

**Recommendation:** Accept
**Confidence:** Certain

**Metareview:**

This paper presents a method for blind separation of correlated sources, which is a challenging task. Applying the weight similarity matching approach to the Det-Max optimization, the authors develop a biologically-plausible two-layered neural network that can separate correlated sources from their linear mixture. All of reviewers agree that the paper is well written and has a solid contribution in BSS. The approach in this paper is a general framework that applies to various source domains. A downside is in experiments on real-world data, which has been improved during the author rebuttal period. Therefore, I am pleased to suggest the paper to be accepted.


**Award:**

No

---

### Decision · Program_Chairs · 2022-09-14

Accept